# Comparative Study of Measurement Methods for Embedded Bioimpedance Spectroscopy Systems

**DOI:** 10.3390/s22155801

**Published:** 2022-08-03

**Authors:** Bilel Ben Atitallah, Ahmed Yahia Kallel, Dhouha Bouchaala, Nabil Derbel, Olfa Kanoun

**Affiliations:** 1Faculty of Electrical Engineering and Information Technology, Chemnitz University of Technology, 09111 Chemnitz, Germany; ahmed-yahia.kallel@etit.tu-chemnitz.de (A.Y.K.); olfa.kanoun@etit.tu-chemnitz.de (O.K.); 2National Engineering School of Sfax, University of Sfax, Sfax 3038, Tunisia; bouchaala.dhouha@gmail.com (D.B.); nabil.derbel@enis.tn (N.D.)

**Keywords:** bioimpedance spectroscopy, bioimpedance, FFT, GPD, IQ demodulation, embedded impedance spectroscopy

## Abstract

Bioimpedance spectroscopy (BIS) is an advanced measurement method for providing information on impedance changes at several frequencies by injecting a low current into a device under test and analyzing the response voltage. Several methods have been elaborated for BIS measurement, calculating impedance with a gain phase detector (GPD), IQ demodulation, and fast Fourier transform (FFT). Although the measurement method has a big influence on the measurement system performance, a systematical comparative study has not been performed yet. In this paper, we compare them based on simulations and experimental studies. To maintain similar conditions in the implementation of all methods, we use the same signal generator followed by a voltage-controlled current source (VCCS) as a signal generator. For performance analysis, three DUTs have been designed to imitate the typical behavior of biological tissues. A laboratory impedance analyzer is used as a reference. The comparison addresses magnitude measurement accuracy, phase measurement accuracy, signal processing, hardware complexity, and power consumption. The result shows that the FFT-based system excels with high accuracy for amplitude and phase measurement while providing the lowest hardware complexity, and power consumption, but it needs a much higher software complexity.

## 1. Introduction

Impedance spectroscopy is an established measurement method for characterizing electrical properties, i.e., complex conductivity and permittivity of materials and systems [1]. Impedance measurement systems have been developed for a variety of applications, including material characterization with inductive sensors [2], chemical parameters investigation [3], and characterization of biological tissues [4,5]. Impedance spectroscopy can be applied to monitor human activity [6], diagnose muscles [7], detect and characterize cancer [8], and monitor human health on a daily basis [9]. In this case, it is called bioelectrical impedance spectroscopy (BIS) [5], which is the main focus of this paper. BIS is a non-invasive technique that injects a constant safe current into the biological tissue or human body through a pair of electrodes in a predefined frequency range. Then, a voltage is sensed through the second pair of electrodes reflecting the actual status of the biological tissues, and the bioimpedance is calculated based on the analysis of these two signals [10].

Unlike other device-under-test (DUT), bioimpedance spectroscopy is injected into human tissues, which requires further attention during system design. It is crucial to design a system that is resilient to the signals generated by the human body while safe enough for human injection. Conventionally, large laboratory equipment has always been used. Nevertheless, such devices are typically bulky and require a permanent infrastructure. Nowadays, those devices are phasing out and being replaced with bioimpedance-embedded solutions. These portable devices are gaining more accessibility owing to their minimal costs, weight, and size. Not only that but many of them can be wearable, which makes them more practical, especially in the era of continuous data collection for machine learning applications. However, this miniaturization of the bioimpedance device is not without drawbacks. As one important part, the system should consider many aspects, including measurement accuracy, lower power consumption, system size, and cost.

In this direction, several studies are carried out. Santoso et al. [11] developed a BIS system based on a gain phase detector (GPD) chip from analog devices covering the frequency range from 1 kHz to 100 kHz with a maximum deviation of 1.5%. Patil et al. proposed a bioimpedance spectrometer based on a GPD module working in the β dispersion range for total body water monitoring [10]. In the same context, Anusha et al. [12] designed a measurement system based on the same technique and validated it in a load range from 1.6 Ω to 1.6 kΩ in the frequency range from 500 Hz to 1 MHz. On the other hand, a second technique has been investigated based on I-U measurements and fast Fourier transform (FFT) for impedance calculation. Munjal et al. [13] propose a solution with an extended frequency range until 10 MHz using the under-sampling technique and achieves a maximal standard deviation of 0.5% and 0.55° deviation in amplitude and phase relatively. Based on the same technique, Vergas et al. designed a portable system for detecting abnormalities in the gastrointestinal tract in the range from 10 Hz to 1 MHz with an accuracy of 2% [14]. A third technique is implemented for BIS systems IQ demodulation measurements. Ibrahim et al. proposed a measurement solution for impedance range from 1–120 Ω for a frequency range from 4 kHz to 120 kHz [15]. Huynh et al. developed a measurement solution using the IQ demodulation technique to monitor blood pressure based on bio-impedance measurements [16]. In the same direction, based on the IQ demodulation technique, Hernánder et al. proposed a completed BIS device for body composition estimation with a sophisticated calibration method for better measurement accuracy [17]. From another perspective, Kusche et al. evaluate the potential of BIS measurement for hand gestures recognition based on novel extracted features [6].

Based on the research on concurrent BIS systems and the literature study, three primary techniques are mainly used for impedance measurements: gain phase detector, I-U measurement based on fast Fourier transform analysis, and I-Q demodulation measurements. However, a considerable divergence in the literature can be noted. For example, many research papers do not compare the method performance to reference measurement but use the standard deviations between the measurements. While this could be useful to demonstrate the precision of such devices, no information about the accuracy can be extracted. Other papers do not compare these methods fairly as separate excitation, and measurement parts are used in the comparison. All this diversity allows only the comparison on the system level but not on the method level, which is more interesting. Accordingly, this paper focuses on studying and evaluating the three measurement techniques under the same conditions to conclude the advantages and limitations of each one for further development. For that purpose, as a first step, a signal generation unit based on AD9850 commercial direct digital synthesis (DDS) is implemented for all measurement methods, followed by a voltage-controlled current source (VCCS) Tietze cascade structure as they are the shared blocks during the comparison. The excitation signal is typically a stepped sine ranging from 1 kHz to 1 MHz to cover β and α dispersion. Then each measurement technique is implemented separately and evaluated using three selected DUTs: RC parallel, Cole–Cole model 1, and Cole–Cole model 2. Finally, the comparative study is carried out based on the impedance measurement, phase and magnitude relative deviations, power consumption, and hardware and signal processing complexities.

## 2. Overview of Impedance Determination Methods

In this section, an overview is provided of the different analysis methods for AC signals to determine the impedance spectrum.

### 2.1. Gain Phase Detector (GPD)

The complex impedance can be calculated by determining the magnitude and phase of a DUT. The magnitude detection is typically done using cascaded linear amplifiers to detect the peak values, which act similarly to an envelope detector. On the other hand, the phase detection is done with a phase-locked loop’s help. An example of gain phase detectors is Analog Devices’s AD8302 chip, typically used for measuring gain and phase in numerous communication and instrumentation applications.

The AD8302 has mainly two input signals, which are logarithmically compressed by the two closely matched logarithmic amplifiers, and the ratio of the difference to the input signal is determined by a subtractor. The AD8302 supports a wide range of frequencies up to 2700 MHz providing accurate measurements within an amplitude ratio of ±30 dB of the two signals. Simply, the AD8302 is comparing the two input voltages $V_{INPA}$ and $V_{INPB}$ and outputs two voltage signals $V_{MAG}$ and $V_{PHS}$ with a dynamic range of 0 to 1.8 V.

In Figure 1, the relationship between input signals ($V_{INPA}$ and $V_{INPB}$) and output signals ($V_{MAG}$ and $V_{PHS}$) is illustrated. With the AD8302, the magnitude measurement range is from 30 dB to +30 dB at a sensitivity of 30 mV/dB, and the phase measurement range is (180 to 0) degrees at a sensitivity of 10 mV/degree or (0 to 180) degrees at a sensitivity of 10 mV/degree. Based on the characterization diagram, the mathematical formulas are as follows:(1)$$Mag\left(\Omega \right)=10^{\frac{V_{MAG}-900\;\mathrm{mV}}{600\;\mathrm{mV}}}$$
(2)$$\theta {(}^{\circ })=\theta_{INPA}-\theta_{INPB}=\pm (\frac{900\;\mathrm{mV}-V_{PHS}}{10\;\mathrm{mV}}+90^{\circ })$$

The real and imaginary parts of the complex impedance can then be described as:(3)$$\left|Z\right|=Mag\cdot R_{shunt}$$
(4)$$\bar{Z}=\left|Z\right|∠\theta$$

Finally, an analog-to-digital converter (ADC) is used to read and transfer the voltage levels to the embedded microcontroller.

### 2.2. IQ Demodulation

The in-phase/quadrature-phase (IQ) demodulation technique exploits the fact that sinusoidal signals with initial phases can be expressed as the sum of sine and cosine without phase shifts [18]. In this direction, the real and imaginary parts, i.e., amplitude and phase, can be extracted by extracting the cosine and sine parts of the signal. The architecture of the in-phase/quadrature-phase (I/Q) demodulation measurement technique is shown in Figure 2.

By injecting a current signal from the VCCS and passing through the impedance interface, an AC current is obtained, and a complex voltage signal ($V_{in}$) is measured at the output of the impedance interface according to Ohm’s law, as follows:(5)$$V_{in}=I\cos\left(2\pi f\right)\left|Z\right|∠\theta =I\left|Z\right|\cos\left(2\pi f+\theta \right)$$

The phase and amplitude of the voltage signal provide the necessary information about the real and imaginary parts of the impedance. In this architecture, the voltage across the DUT is measured with an instrumentation amplifier. Compared to other measurement techniques, no shunt resistor is required in the circuit structure. To extract the real and imaginary parts of the impedance, $V_{in}$ is amplified by a gain factor *k* and then multiplied by two local oscillatory signals of the same frequency, which are the in-phase (I) and quadrature-phase (Q). The I and Q components of the signal are calculated as in Equations (Equation 6) and (Equation 7).
(6)$$V_{i}=kV_{i}V_{iLO}\left(t\right)=\frac{1}{2}kI\left|Z\right|\left[\cos\theta +\cos\left(4\pi f+\theta \right)\right]$$
(7)$$V_{q}=kV_{i}V_{qLO}\left(t\right)=\frac{1}{2}kI\left|Z\right|\left[\sin\theta -\sin\left(4\pi f+\theta \right)\right]$$

The previous equations show that the amplitude and phase information are found on the DC voltage. Low pass filters (LPFs) can therefore be designed to filter out all other frequencies of $V_{i}$ and $V_{q}$, leaving only the DC components. The extracted DC components provide the impedance’s real and imaginary components. In the ideal case, the complex signals after filtering by the LPFs are:(8)$$V_{i}=\frac{1}{2}kI\left|Z\right|\cos\theta \Rightarrow Z_{r}=\frac{2}{kI}\cos\theta$$
(9)$$V_{q}=\frac{1}{2}kI\left|Z\right|\sin\theta \Rightarrow Z_{i}=\frac{2}{kI}\sin\theta$$

Finally, an ADC is used to convert the voltage into a digital quantity that microcontrollers can interpret.

### 2.3. Impedance Measurements Based on Fast Fourier Transform (FFT) and I-U Technique

Discrete Fourier transform (DFT) is a tool to calculate the amplitude and phase spectrum of a signal. Unlike discrete-time Fourier transform (DTFT), DFT operates on a finite number of samples and frequencies. By selecting the frequencies of the excitation signal, DFT can be used to determine the voltage and current amplitude and phase at the given frequency. After that, the impedance $\underline{Z}$ can then be expressed as a function of the voltage $\underline{V}$ and current $\underline{I}$ as follows:(10)$$\underline{Z}=\frac{\underline{V}}{\underline{I}}$$

In practice, the impedance measurement could be realized based on the I-U measuring method, which is based on the analysis of two signals crossing the DUT and the shunt resistor, respectively, as illustrated in Figure 3:

The impedance is calculated based on Ohm’s law, which is mainly the ratio of voltage amplitude by the current amplitude, which is explained as follows:(11)$$Z_{DUT}=\left(\frac{U_{Voltage}}{U_{Current}}\right)Z_{Shunt}$$

Furthermore, the phase impedance is calculated from the difference of the individual calculation of voltage and current phases, respectively, as demonstrated below:(12)$$\theta =\theta_{Voltage}-\theta_{Current}=\tan^{-1}\left(\frac{V_{imaginary}}{V_{real}}\right)-\tan^{-1}\left(\frac{I_{imaginary}}{I_{real}}\right)$$

The real and imaginary components are the results of the signal processing using the FFT algorithm, which is the hardware-optimized version of DFT. Given *N* samples at a sampling frequency $F_{s}$, and at a frequency index *k* corresponding to frequency $f=k\frac{Fs}{N}$, the real and imaginary components are calculated as follows:(13)$$X\left[k\right]=\sum_{n=0}^{N-1}x\left(n\right)\left[\cos\left(\frac{2\pi nk}{N}\right)-j\sin\left(\frac{2\pi nk}{N}\right)\right]=X{\left[k\right]}_{real}+jX{\left[k\right]}_{imaginary}$$
where *j* is the imaginary unit.

## 3. Implementation and Evaluation

In this section, we propose to compare and validate the proposed method experimentally. For this purpose, a galvanostatic architecture has been deployed to evaluate the different impedance determination methods. A Tietze-based VCCS has been used for the current injection. The VCCS is fed by a waveform generator based on the direct digital synthesis method. A measurement circuit based on the proposed method is used to evaluate the impedance values.

### 3.1. Voltage Generator

An AD9850 module from Analog Devices is used to generate the AC voltage. The AD9850 is a highly integrated signal generator that combines modern DDS technology with an inbuilt high-speed, high-performance D/A converter and comparator to build a fully-featured, digitally programmable frequency synthesizer and clock generator. With a 32-bit frequency tuning word and a 125 MHz reference clock input, the AD9850’s DDS achieves an output tuning resolution of 0.0291 Hz and can reach a maximum of 40 MHz. An STM32 micro-controller controls the module through an SPI communication protocol to set the frequency and phase of the generated signals. The module’s output signal is purely positive, as shown in Figure 4. First, a passive high-pass filter of the first order is implemented with a voltage follower structure to avoid interference between circuit blocks to remove the DC component and filter the unnecessary low frequencies. Afterward, a non-inverting voltage amplifier structure is implemented to amplify the output voltage to 4 volts peak to peak as an input to the VCCS structure as it is required.

### 3.2. Tietze Cascade VCCS

As a current signal injection necessary for bioimpedance spectroscopy, a Tietze cascade VCCS has been deployed within the measurement system. The impedance measurement system must satisfy specific technological and medical safety regulations. During measurement, the current or the voltage will change proportionally depending on the injection type, i.e., a voltage or current. As per human safety regulation, the current must be maintained at sub-mA levels. For this reason, a galvanostatic architecture realized through a voltage-controlled current source is used. The VCCS aims to maintain a constant AC amplitude regardless of signal frequency or load impedance [19]. This paper investigates a Tietze cascade topology based on two cascaded operational amplifiers.

In order to assure a high bandwidth operation, i.e., no drop due to frequency change, Analog Devices AD8021, which is a high-bandwidth operational amplifier, is selected. AD8021 offers a bandwidth of 490 MHz and a common-mode rejection ratio (CMRR) of 98 dB, making it very suitable for bioimpedance spectroscopy.

The transfer function between the output current and the input voltage can be derived from the amplifier calculation principle as follows:(14)$$i_{out}=-\frac{R_{2}}{R_{1}}\frac{2R_{6}+R_{3}+R_{4}+2R_{5}}{R_{6}\left(R_{3}+R_{4}\right)-R_{2}\left(R_{3}+R_{4}+2R_{5}\right)}V_{in}$$
The balance condition is as follows:(15)$$R_{1}=R_{2}=R_{3}=R_{4}=R_{5}+R_{6}$$
which, once fulfilled, results in the following current:(16)$$i_{out}=\frac{1}{R_{5}}V_{in}$$

The Tietze cascade VCCS structure is first simulated using LTspice software to ensure correct delivery performance. In addition, the spice amplifier model offered by the manufacturer is used, which considers the parasitic effects based on the manufacturer’s evaluations. Then, it is implemented on a PCB. Figure 5 shows the circuit architecture used in this paper. In order to evaluate the bandwidth performance of the VCCS and also the impedance response, the VCCS structure was tested under the following resistance loads: 560 Ω, 1 kΩ, 1.5 kΩ, 2.2 kΩ, 2.7 kΩ, 3.3 kΩ, 3.9 kΩ, 4.7 kΩ, 5.6 kΩ, and 6.8 kΩ and within a frequency range from 1 kHz to 1 MHz. Figure 6 shows the recorded VCCS output at 10 kHz of four loads 1.5 kΩ, 2.7 kΩ, 4.7 kΩ, and 6.8 kΩ successively. Comparing the transient analysis in simulation (Figure 6a) and experimentation (Figure 6b), it can be concluded that the performance of VCCS is stable and is compliant with the expectations.

In a second step, an AC analysis is carried out of the same loads’ values, the frequency range from 1 kHz to 1 MHz. Here, a 1 V peak to peak is used as an input voltage for the VCCS structure. Based on Equation (Equation 16), the output current amplitude is expected to be 200 μA. As illustrated in Figure 6c, the simulation results show that VCCS is stable and the delivered current is constant throughout the frequencies range of interest, which conforms with the theoretical value. However, the experimentation results in Figure 6d show that the VCCS is only stable until 200 kHz. At 200 kHz, a logarithmic current value drop is visible. The drop offset and steepness also depend on the load values. This drop is attributed to the non-ideality of the electronic components in a practical implementation e.g., amplifier characteristics and passive components tolerances. In practice, the frequency response of the Tietze VCCS shows similar behavior to a low-pass filter, which is due to the bandwidth limitations of the VCCS structure in practice.

### 3.3. Measurement Circuits

As a next step to validate the proposed system, the measurement circuit block is examined and evaluated compared to the simulation, as explained in Section 2. GPD and FFT-based I-U methods use the same measurement structure as they both measure sinusoidal signals referenced to the voltage difference potential across the DUT and the voltage signal across the shunt resistor for current measurement (see Figure 7a). The only difference in the FFT-based I-U method is that an offset voltage of 1.5 V has been added to ensure that the voltages remain positive so that the STM32 ADC can read them. However, this is not required for the GPD system. The output of the IA amplifier for a complex DUT (RC parallel) is recorded and compared to the simulated signals as shown in Figure 7b,c. No visible distortion of the signals is detected. The frequency is stable, and the voltage amplitude is comparable to the simulation.

On the other hand, the measurement circuit of the IQ demodulation method is more complex, as shown in Figure 8a. As explained in Section 2, this method uses only one instrumentation amplifier to measure the differential potential across the DUT. Then this signal will be multiplied with an in-phase and quadrature-phase signal using the AD835 chip. Finally, a low-pass filter is implemented as the last step to get the relative DC voltages to the real and imaginary components.

For the choice of an adequate low-pass filter, a comparative simulation study between four low-pass filters is carried out: second and first-order passive LPF, Sallen-key, and Rauch structures. The Sallen-Key filter structure shows better performances in terms of convergence time, magnitude, and phase deviations. Therefore, this filter is chosen and implemented. Furthermore, this filter is multiplexed and is used sequentially for the in-phase and quadrature signals. Finally, the output of the multiplier and filter output for each I-Q component is compared to the simulation as shown in Figure 8b,c. The analog system is showing similar behavior to the simulation with slight deviations due to the non-ideality of components.

## 4. Results and Discussion

The three measurement techniques are implemented on separate PCBs. The PCBs maintain the same designs as the standard electronic blocks to avoid any possible influences on PCB layout. The techniques will be tested on three selected DUTs for this work: R/ /C element, Cole–Cole model 1, and Cole–Cole model 2. In the experimental study, all the DUTs were measured using the commercial impedance analyzer Agilent 4294a as a reference. The measurements are done on 100 frequency points from 1 kHz to 1 MHz for the GPD and FFT-based systems and only for 30 frequency points for the IQ demodulation system, which belongs to the 100 main points. This limitation in the latter method is mainly due to the limited standard values of resistor and capacitors values required to build a local oscillator circuit for the generation of 90° phase shift signal.

### 4.1. RC Elements

As the first evaluation, a 4.7 kΩ in parallel with a capacitor of 2.2 nF is taken as DUT. Figure 9 illustrates the archived results. The GPD-based system in Figure 9a,b are shown an enormous deviation in the magnitude and phase measurement, which are about 38% and 11° at low frequencies, respectively. This is mainly due to the over-range voltage in the INPA pin in AD8302, which exceeds the maximum allowed value of 0.7 V. This explains the considerable measurement deviation and shows the major limitation of this technique. The GPD-based system achieves a mean deviation of 33.25% and 3.437° for the magnitude and phase, respectively. On the other hand, the IQ demodulation system is showing better results with a mean deviation of magnitude relative deviation equal to 8.593% for the magnitude measurements and 3.394° for phase measurements. The FFT-based system presents the highest accurate results with 1.122% as the mean relative magnitude deviation and 0.972° mean of absolute phase deviation.

### 4.2. Cole–Cole Model

A Cole–Cole model is one most accessible equivalent models of biological tissue. As shown in Figure 10a, a 330 Ω resistor is in series to parallel 1.5 kΩ and 10 nF. These values are selected to simulate the real body composition of extracellular liquid resistance, intracellular liquid resistance, and membrane capacitance, respectively. The output of the instrumentation amplifiers is ensured to be less than 0.7 V for all frequencies to overcome the limitation of the GPD system and ensure the same comparison condition. The GPD-based system performs much better than the previous RC parallel, where the deviation decreased to 9.1% and 3.243° for the magnitude and phase, respectively. The IQ demodulation achieves a 3.299% relative mean deviation of magnitude and 1.803° mean deviation of phase. The IQ-based system shows higher deviation at higher frequencies due to the low voltage output of the low-pass filter. It can be remarked that the output decreases down to a few mV, which is at the proximity of ADC quantization level and noise, which lowers the measurement accuracy at that range.

On the other hand, the UI/FFT-based system shows accurate results, reaching only 1.643% mean relative deviation in magnitude and 0.751° mean relative deviation of phase. Nonetheless, this system shows a higher deviation in higher frequencies in the proximity of the sampling frequency. The used STM32 μC has a sampling frequency of 1.8 MHz, and the higher the frequencies, the more quantization deviation is expected [18]. Nevertheless, this sampling frequency is sufficient to obtain high accuracy, especially in phase measurements until 200 kHz, which is adequate for the targeted application.

A second possible form of the Cole–Cole model (see Figure 11) is tested and evaluated with the same passive components. All system shows comparable performance to the first Cole–Cole model. The mean relative deviations of the magnitude are 8.159%, 5.145%, and 1.948% for the GPD, IQ, and FFT-based systems, respectively. On the other side, the mean relative deviations of phase 2.605°, 1.661°, and 0.886°, respectively.

As a partial conclusion, the UI/FFT-based system has better accuracy in terms of magnitude and phases compared to GPD and IQ for impedance measurements in the same range as the Cole–Cole model.

### 4.3. Comparative Study

Three systems for BIS based on different measurement techniques are well implemented and tested under the same conditions. The systems are tested in the frequency range from 1 kHz to 1 MHz using the same signal generator, and VCCS modules explained above. Based on the realized experiments, the accuracy in term phase and magnitude measurements are evaluated. Table 1 summarizes the achieved results while measuring three different DUTs (RC parallel, Cole–Cole model type 1, and Cole–Cole model type 2). However, the aim is not to compare only the system performance but also the system design challenges and requirements, e.g., signal processing and hardware complexities and power consumption, as shown in Figure 12. The FFT-based measurement system shows accurate results, with a maximum deviation of 1.94% for relative magnitude and 0.972° for phase deviation to reference measurement from the 4294a Agilent impedance analyzer. In addition, the analog measurement structure is less complex, containing a total of 11 components, including only two integration circuits (ICs). However, this method requires more complex signal processing than other methods, as several aspects must be considered, such as sampling frequency and spectra leakage effect. For that reason, a robust μC is needed for embedded systems like STM32 to ensure accurate measurements for higher frequencies, which is also challenging.

On the other hand, the GPD solution has less signal processing complexity as the AD8302 GPD module provides two relative DC voltages to the magnitude and phase, respectively, with a not complex analog structure containing three main ICs and 25 components in total. The magnitude and phase will be calculated based on predefined equations given by the manufacturer. Nevertheless, the GPD system is highly noise-sensitive, and the DC output can be easily affected. In addition, the main limitation of this solution is that the input sine voltages should be less than ±0.7 V. Otherwise, the system cannot measure the impedance correctly, which leads to massive deviations as the case with the tested RC parallel as mentioned in Table 1 with a mean deviation of 33.25% for the magnitude.

However, the GPD system shows acceptable results with phase measurement where the mean deviation is around 3°. The results from the IQ demodulation are more accurate than the GPD results, with around 3.2% relative deviation of the magnitude and 1.8° of relative phase deviation for the Cole–Cole model. This method also does not require a high signal processing complexity as the DC filter outputs will be used directly to calculate the real and imaginary parts of the complex impedance. On the other hand, this technique requires a highly complex analog structure due to the convolution process and filtering stage with 30 components, including six ICs. This explains the high power consumption of this technique, which is approximately 290.5 mW compared to the GPD and UI techniques, with power consumption equal to 159.2 mW and 122.4 mW, respectively.

As a perspective, the FFT-based measurement solution has the potential to support and analyze a multifrequency excitation signal compared to the IQ and GPD systems, which can ensure faster measurements for fast real-time application requirements.

## 5. Conclusions

BIS is a widely utilized measurement technique in a variety of research fields. This paper presents a systematical comparative study of possible BIS measurement methods under well-defined conditions. For that purpose, three bioimpedance measurement methods, namely, the gain phase detector, I/Q demodulation, and FFT-based system, have been investigated, implemented, and compared regarding amplitude accuracy, phase accuracy, hardware and computational complexity, frequency range, and power consumption. For the evaluation of proposed systems, a common signal generator module based on AD9850 and a Tietze cascade structure acting as a voltage-controlled current source have been used as excitation blocks. Then, the developed solutions were tested using the different DUTs (RC parallel and Cole–Cole models) that mimic the typical biological tissue’s behavior in the frequency range from 1 kHz to 1 MHz. The experimental investigations show that the FFT-based method achieves the most accurate results with a mean relative deviation of 1.6% for the magnitude and 0.751° for the Cole–Cole model type 1. The IQ and GPD systems achieve 5.1% and 8.1% mean of relative magnitude deviation and 1.661° and 2.605° mean of phase deviation respectively. In addition, the FFT-based system requires less hardware complexity with less number of components, which explains its low power consumption of 122.4 mW compared to the IQ and GPD solutions with 290.5 mW and 159.2 mW respectively. On the other hand, this measurement method requires the most complex signal processing to avoid spectral leakage and to carry out impedance calculations. This study can be applied to various impedance spectroscopy applications with comparable impedance and frequency ranges even though it primarily targets bioimpedance spectroscopy.

## Figures and Tables

**Figure 1 sensors-22-05801-f001:**
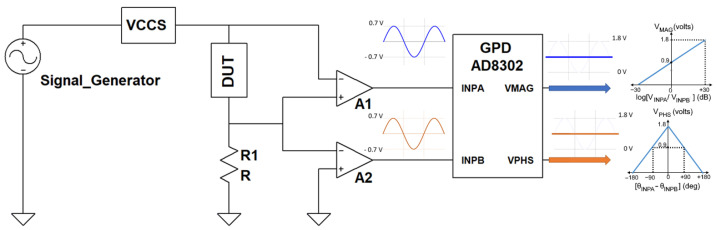
Gain phase detector measurement technique using AD8302.

**Figure 2 sensors-22-05801-f002:**
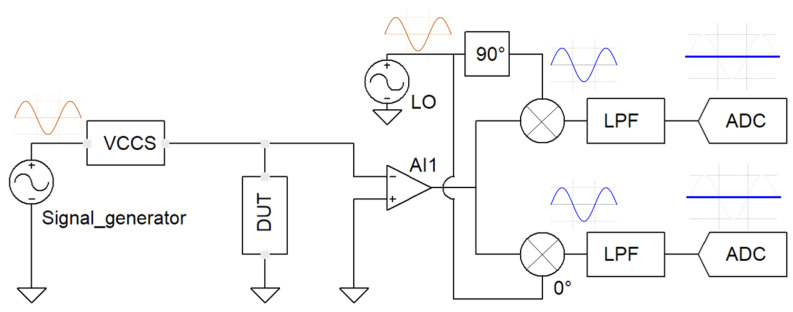
I/Q demodulation measurement technique.

**Figure 3 sensors-22-05801-f003:**
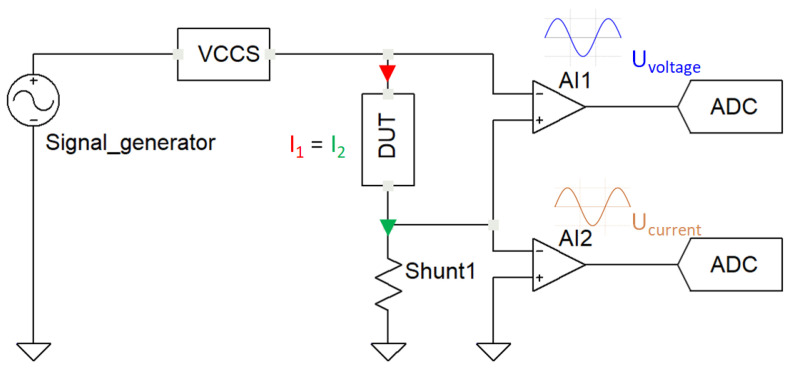
UI measurement method.

**Figure 4 sensors-22-05801-f004:**
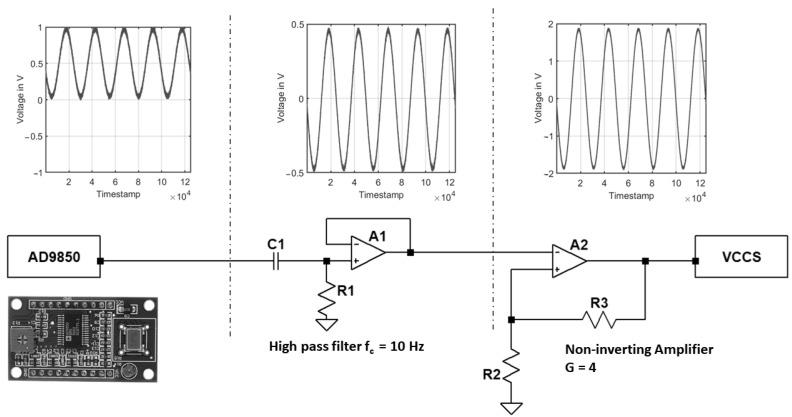
Signal generator module based on AD9850 module.

**Figure 5 sensors-22-05801-f005:**
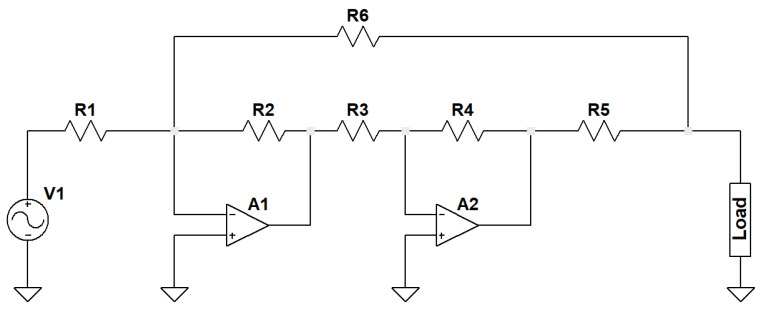
Tietze cascade VCCS structure.

**Figure 6 sensors-22-05801-f006:**
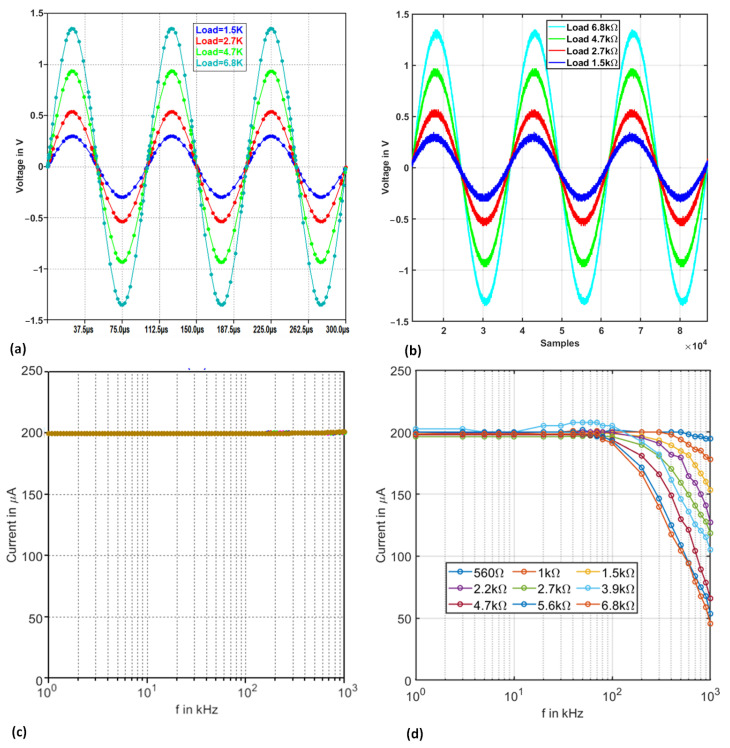
VCCS analysis: Output signal cross the load for transient analysis: (**a**) simulation results, (**b**) implementation results, AC analysis, (**c**) simulation results, and (**d**) implementation results.

**Figure 7 sensors-22-05801-f007:**
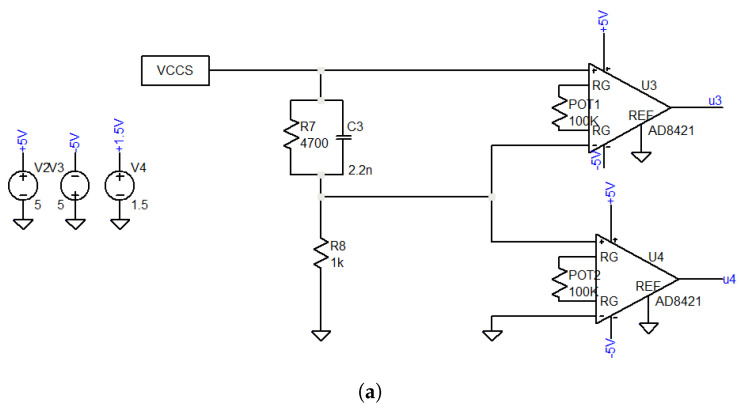
I-U measurement module: (**a**) sensing structure (**b**) simulation results, and (**c**) implementation results.

**Figure 8 sensors-22-05801-f008:**
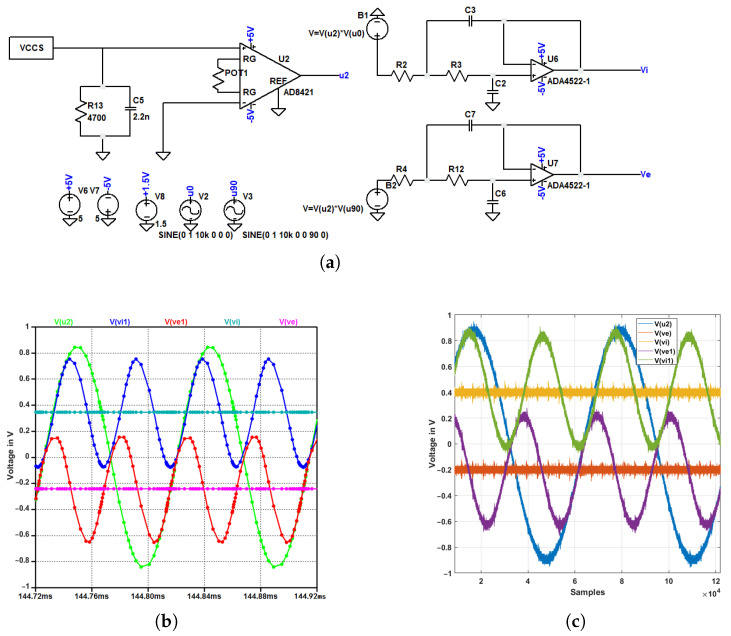
IQ demodulation measurement module: (**a**) sensing structure (**b**) simulation results, and (**c**) implementation results.

**Figure 9 sensors-22-05801-f009:**
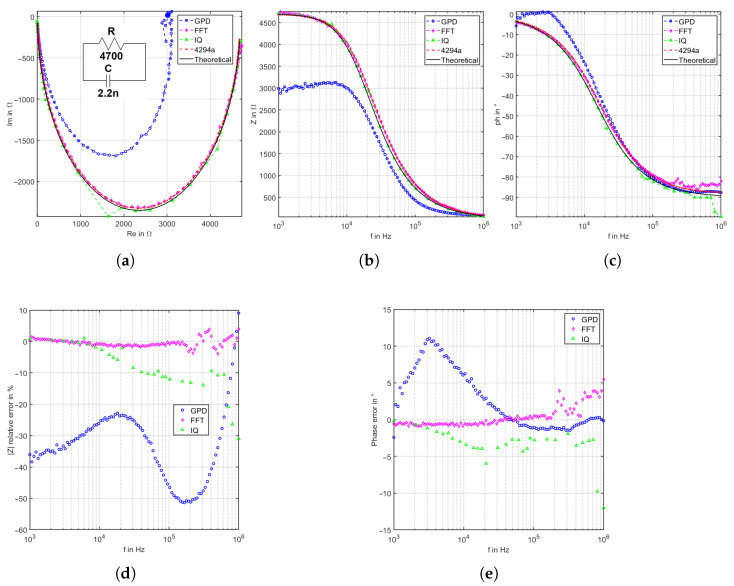
Results for the RC element (4.7 kΩ and 2.2 nF): (**a**) Nyquist plot, (**b**) magnitude bode plot, (**c**) phase bode plot, (**d**) magnitude relative deviation, and (**e**) phase relative deviation.

**Figure 10 sensors-22-05801-f010:**
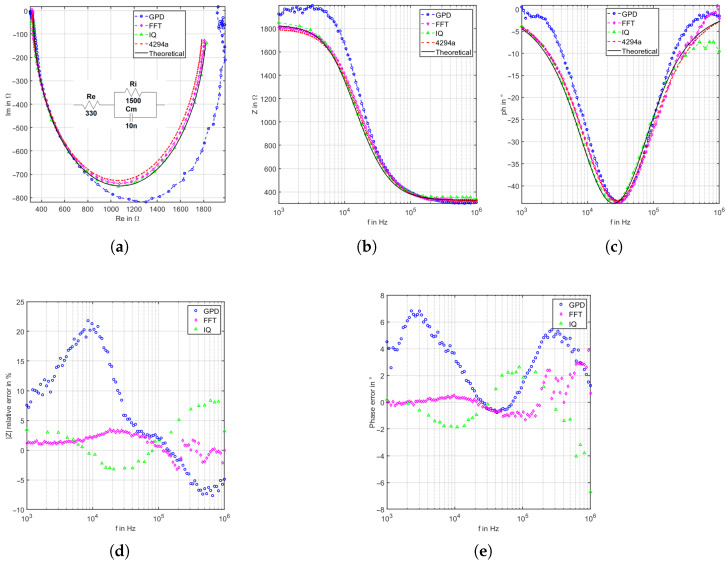
Evaluation of Cole–Cole model 1 using 330 Ω resistor in series to parallel 1.5 kΩ and 10 nF. (**a**) Nyquist plot, (**b**) magnitude bode plot, (**c**) phase bode plot, (**d**) magnitude relative deviation, and (**e**) phase relative deviation.

**Figure 11 sensors-22-05801-f011:**
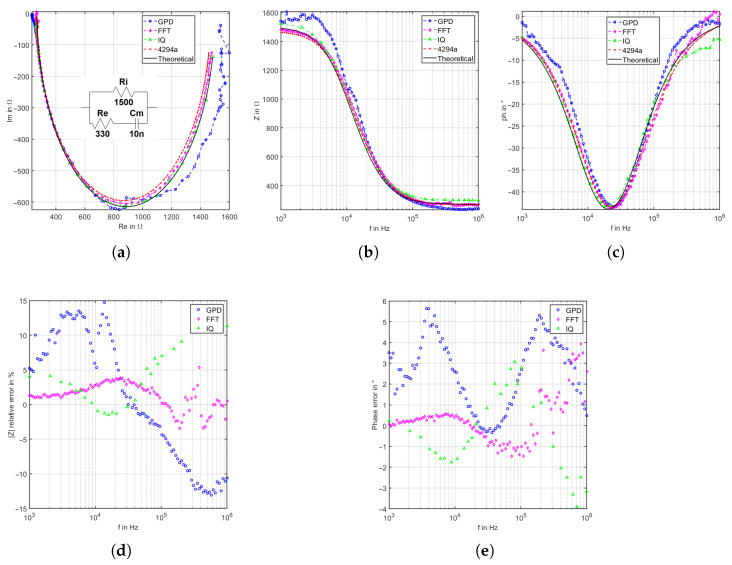
Evaluation of Cole–Cole model 2 using 1.5 kΩ parallel to 330 Ω resistor in series to 10 nF. (**a**) Nyquist plot, (**b**) magnitude bode plot, (**c**) phase bode plot, (**d**) magnitude relative deviation, and (**e**) phase relative deviation.

**Figure 12 sensors-22-05801-f012:**
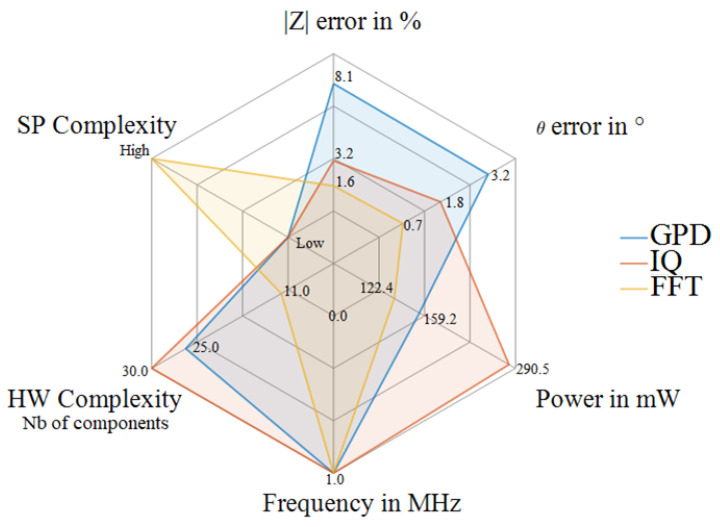
Comparative plot of developed systems.

**Table 1 sensors-22-05801-t001:** Comparative study and deviation evaluation of tested DUTs.

	Mean of Modulus Relative Deviation in %			Mean of Phase Deviation in Degree		
**DUT**	**GPD**	**FFT**	**IQ**	**GPD**	**FFT**	**IQ**
RC element	33.251	1.122	5.593	3.437	0.972	3.394
Cole–Cole model 1	9.126	1.643	3.299	3.243	0.751	1.803
Cole–Cole model 2	8.159	1.948	5.145	2.605	0.886	1.661

## Data Availability

Not applicable.

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
