# Peer review of "Comparative Study of Measurement Methods for Embedded Bioimpedance Spectroscopy Systems"

_sensors, 2022, doi:10.3390/s22155801_

Round 1

Reviewer 1 Report

I recommend some improvements before its publication:

·      Page 2 line 73-77, explanation of some abbreviation is not given. It can be difficult to follow for readers. Same for page 3 line 95-96. Authors should check the whole manuscript.

·      Some abbreviations are given more than one time, ex: (VCCS)

·      Figure 6a: Low quality image.

·      Sometimes a section lacks a main thesis sentence to start and a summary to end, which makes it hard to follow. 

·      Conclusion part should be rewritten. It is not clear enough.

·    The recent studies discovered are not clearly introduced.  Authors should add more recent studies to support their study. 

Reviewer 2 Report

Review on: Comparative Study of Measurement Methods for Embedded Bioimpedance Spectroskopy Systems (sensors - 1809996)

The work described in this manuscript compares three different measurement methods for bioimpedance spectroscopy. The manuscript is well structured and includes a sufficient introduction, an overview of established impedance measurement methods with theoretical considerations, a clear description of the experimental work as well as a discussion of the results and a conclusion.

The following is a list of comments for further improvement of the paper:

Line 9 (Abstract): typing error “signla generator”

Line 34: there is an unfinished sentence: “Conventionally, large laboratory equipment.”

Line 123: here is e reference to equation 9 and 8. Please check the correctness. Possibly, it should refer to equation 6 and 7

Line 198: “As illustrated in Fig. 6d c, …” do the authors mean Fig. 6a and 6c? The figure caption indicates, that simulation results are shown in Fig. 6a and 6c.

Line 202: “This drop is attributed to the non-ideality of the electronic components …” The authors should add some information about the component models used in the simulation. In the simulation environment, there are ideal models and models with more realistic characteristics (i.e. with parasitic resistances, parasitic capacitances and parasitic inductors) available. What exactly causes the drop in Fig. 6d?

Line 273: Reference in square brackets is missing
